# Complement 3a receptor 1 on macrophages and Kupffer cells is not required for the pathogenesis of metabolic dysfunction-associated steatotic liver disease

Edwin A Homan, Ankit Gilani, Alfonso Rubio-Navarro, Maya A Johnson, Odin M Schaepkens, Eric Cortada, Renan Pereira de Lima, Lisa Stoll, James C Lo*

Division of Cardiology, Department of Medicine, Cardiovascular Research Institute, Weill Center for Metabolic Health, Weill Cornell Medicine, New York, United States

## eLife Assessment

This **valuable** study investigates the role of Complement 3a Receptor 1 (C3aR) in the pathogenesis of Metabolic Dysfunction-Associated Steatotic Liver Disease (MASLD) using mouse models with specific target deletions in various cell types. While the general relevance of C3aR in inflammatory contexts has been established before, the authors provide **solid** evidence here that C3aR does not contribute significantly to MASLD pathogenesis in their models. The work will be of interest to colleagues studying diseases of the liver and the intersection with inflammation.

*For correspondence:
jlo@med.cornell.edu

Competing interest: The authors declare that no competing interests exist.

**Abstract** Together with obesity and type 2 diabetes, metabolic dysfunction-associated steatotic liver disease (MASLD) is a growing global epidemic. Activation of the complement system and infiltration of macrophages has been linked to progression of metabolic liver disease. The role of complement receptors in macrophage activation and recruitment in MASLD remains poorly understood. In human and mouse, *C3AR1* in the liver is expressed primarily in Kupffer cells, but is down-regulated in humans with MASLD compared to obese controls. To test the role of complement 3a receptor (C3aR1) on macrophages and liver resident macrophages in MASLD, we generated mice deficient in C3aR1 on all macrophages (C3aR1-MφKO) or specifically in liver Kupffer cells (C3aR1-KpKO) and subjected them to a model of metabolic steatotic liver disease. We show that macrophages account for the vast majority of *C3ar1* expression in the liver. Overall, C3aR1-MφKO and C3aR1-KpKO mice have similar body weight gain without significant alterations in glucose homeostasis, hepatic steatosis and fibrosis, compared to controls on a MASLD-inducing diet. This study demonstrates that C3aR1 deletion in macrophages or Kupffer cells, the predominant liver cell type expressing *C3ar1*, has no significant effect on liver steatosis, inflammation or fibrosis in a dietary MASLD model.

## Introduction

Obesity and related metabolic diseases such as type 2 diabetes (T2D) and metabolic dysfunction-associated steatotic liver disease (MASLD) remain a worldwide epidemic with increasing prevalence (*Ge et al., 2020*; *Younossi et al., 2018*). MASLD describes the constellation of hepatic lipid deposition, inflammation, and fibrosis associated with obesity and T2D that ultimately leads to MASH

cirrhosis, which has become the leading cause of liver transplantation in the United States (*Ferguson and Finck, 2021*; *Friedman et al., 2018*; *Stefan et al., 2019*; *Kim et al., 2021*). Notably, MASLD is increasingly recognized as an important risk-enhancing factor for atherosclerotic cardiovascular disease (*Duell et al., 2022*; *Kasper et al., 2021*).

Liver macrophages help to maintain hepatic homeostasis and consist of embryo-derived resident macrophages called Kupffer cells, which self-renew and do not migrate, or peripheral monocyte-derived macrophages, which infiltrate into liver tissue upon metabolic or toxic liver injury and under certain circumstances can take on Kupffer cell-like identity (*Barreby et al., 2022*; *Cai et al., 2019*; *Guilliams and Scott, 2022*; *Park et al., 2023*; *Sakai et al., 2019*). In obesity, bone-marrow-derived myeloid cells migrate to the steatotic liver, and pro-inflammatory recruited macrophages are postulated to drive the progression of MASLD to MASH (*Krenkel et al., 2020*). Spatial proteogenomics reveals a population of lipid-associated macrophages near bile canaliculi that is induced by local lipid exposure and drives fibrosis in steatotic regions of murine and human liver (*Guilliams et al., 2022*). In addition, deep transcriptomic profiling in human MASLD has identified candidate gene signatures for steatohepatitis and fibrosis with possible therapeutic implications (*Govaere et al., 2020*).

Activation of the body's complement system leads to increased cell lysis, phagocytosis, and inflammation (*Merle et al., 2015*), and it is increasingly recognized as an important contributor to regulation of metabolic disorders such as T2D and MASLD (*Kolev and Kemper, 2017*; *Zhao et al., 2022*). In human liver biopsies, higher lobular inflammation scores correlate with activation of the complement alternative pathway (*Segers et al., 2014*), which can signal *via* the C3a receptor 1 (C3aR1), a $G_i$-coupled G protein-coupled receptor (*Markiewski and Lambris, 2007*). The complement 3 polypeptide (C3) is cleaved by C3 convertase to the activated fragment, C3a, which then binds C3aR1 (*Yadav et al., 2023*). Complement factor D (CFD), also known as the adipokine adipsin, is the rate-limiting step in the alternative pathway of complement activation (*Flier et al., 1987*; *Xu et al., 2001*).

Several studies have reported opposing roles of adipsin and C3aR1 on hepatic steatosis in diet-induced obesity (*Lim et al., 2013*; *Polyzos et al., 2016*; *Han and Zhang, 2021*). Our lab has found that adipsin/CFD is critical for maintaining pancreatic beta cell mass and function (*Lo et al., 2014*; *Gómez-Banoy et al., 2019*). Murine obese and diabetic models such as *db/db* mice and high-fat diet (HFD) feeding result in very low circulating adipsin (*Flier et al., 1987*). Replenishing adipsin in *db/db* mice raises levels of C3a and insulin, lowers blood glucose levels, and inhibits hepatic gluconeogenesis (*Lo et al., 2014*). However, whole-body deletion of C3aR1 decreases macrophage infiltration and activation in adipose tissue, protects from HFD-induced obesity and glucose intolerance, and decreases hepatic steatosis and inflammation (*Mamane et al., 2009*). In a model of fibrosing steatohepatitis, bone-marrow-derived macrophages were found to activate hepatic stellate cells, which was blunted in whole-body C3aR1 KO mice (*Han et al., 2019*).

In the present study, we aim to explore the macrophage-specific effect of complement receptor signaling in MASLD pathogenesis. To determine the consequences of macrophage and Kupffer cell ablation of C3aR1, we use a murine dietary model of MALFD/MASH, the Gubra Amylin Nash (GAN) diet, which has macronutrient similarities to the Western diet and produces similar histologic and transcriptomic changes to human MASLD/MASH (*Boland et al., 2019*; *Hansen et al., 2020*; *Vacca et al., 2024*).

## Results

### C3AR1 is expressed in human and mouse liver, primarily in Kupffer cells

In the scRNA-Seq database, Human Protein Atlas, *C3AR1* is broadly expressed throughout the body, with increased abundance in tissues rich in immunologic cell types, such as bone marrow and appendix (*Figure 1A*; *Uhlén et al., 2015*). In a single-cell transcriptomic database of healthy human liver, *C3AR1* expression predominates in the macrophage and Kupffer cell population, with minimal-to-undetectable *C3AR1* expression in hepatocytes or hepatic stellate cells by scRNA-Seq (*Figure 1B*; *MacParland et al., 2018*). In the mouse liver scRNA-Seq database, Tabula Muris, *C3ar1* is similarly expressed primarily in Kupffer cells (*Figure 1—figure supplement 1A*; *The Tabula Muris Consortium et al., 2018*).

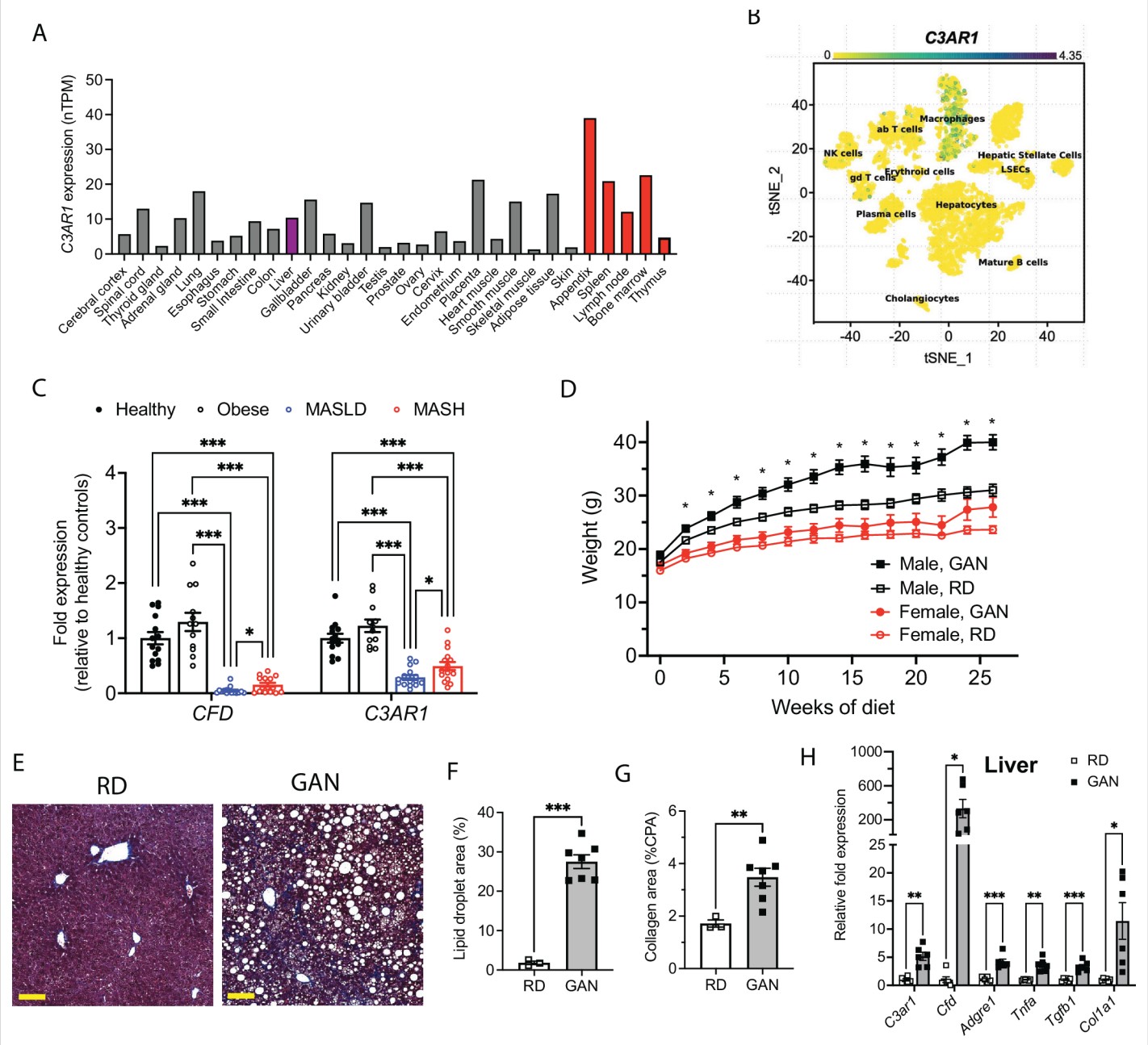

**Figure 1.** C3AR1 is found in macrophages, is modulated by MASLD/MASH in humans, and is induced by a murine dietary model of MASH. (**A**) Relative *C3AR1* human tissue expression level by tissue, derived from deep sequencing of the mRNA combined dataset (HPA and GTEx) in the Human Protein Atlas, shown as normalized transcripts per million (nTPM). Liver is highlighted in purple and immunologic tissues are highlighted in red. (**B**) Single-cell RNA sequencing distribution of *C3AR1* expression in human liver (tSNE, t-distributed Stochastic Neighbor Embedding). (**C**) Analysis of *CFD* and *C3AR1* expression from liver biopsy samples in patients with MASH, MASLD, obesity without MASLD, and age-matched healthy controls (n=12–16 per group, Welch *t* test with Holm-Šídák correction for multiple comparisons). (**D**) Weight curve in male and female *C3ar1^flox/flox^* control mice placed on GAN high-fat diet compared to regular diet (RD) controls (males, n=7; females, n=6). (**E**) Representative liver section staining by Masson's Trichrome in male control mice on RD or GAN diet for 28 weeks (scale bar = 100 mm). (**F**) Lipid droplet area quantification in liver sections from male control mice, excluding vessel lumens (RD, n=3; GAN, n=7). (**G**) Collagen area quantification in liver sections of male control mice (RD, n=3; GAN, n=7). (**H**) Gene expression of key macrophage or fibrosis genes in male control mice on GAN or RD (n=6 per group). Unpaired two-tailed Student's *t* test (Except 1 C as above). Annotations: *, p<0.05; **, p<0.01; ***, <0.001. Error bars represent standard error of the mean.

The online version of this article includes the following source data and figure supplement(s) for figure 1:

**Source data 1.** Source data for *Figure 1A, C, D, F, G and H*.

**Figure supplement 1.** C3ar1 is expressed in liver, primarily in Kupffer cells.

## Hepatic CFD and C3AR1 are downregulated in human MASLD/MASH

We also examined data from Suppli and coworkers, who performed bulk transcriptomic analysis of human liver samples from an age-matched cohort of healthy controls and obese controls without MASLD, as well as MASLD and MASH patients without cirrhosis (*Suppli et al., 2019*). Both *CFD and C3AR1* were unchanged in obese subjects without MASLD compared to healthy controls, but both *CFD* and *C3AR1* were significantly downregulated in liver biopsies from both MASLD and MASH patients compared to both healthy controls and obese subjects without MASLD (*Figure 1C*). Interestingly, both *CFD* and *C3AR1* levels were slightly higher in MASH individuals compared to those with MASLD only.

## Murine MASH model recapitulates key features of human MASH

At 5 weeks of age, we subjected *C3ar1*^*flox/flox*^ control mice to standard regular diet (RD) or GAN diet (*Boland et al., 2019*; *Hansen et al., 2020*). After 28 weeks of GAN diet, male mice gained body weight compared to RD (*Figure 1D*), primarily as fat mass (*Figure 1—figure supplement 1B–C*), but weight gain in female GAN-fed mice was attenuated. Histologic signs of MASLD were present in GAN-fed mice (*Figure 1E*), most notably hepatic steatosis and hepatocyte ballooning (*Figure 1F*), and liver fibrosis measured by collagen deposition nearly doubled with GAN compared to RD (*Figure 1G*). Both hepatic *C3ar1* and *Cfd* gene expression were robustly increased on GAN compared to RD, as were markers of macrophage infiltration, hepatic inflammation, and fibrosis, including collagen gene expression, indicating progression to fibrotic MASH (*Figure 1H*). In female control mice on GAN diet, there were no significant differences in *C3ar1* expression or other gene markers, although there was a nonsignificant trend toward increased inflammation and fibrosis compared to regular diet (*Figure 1— figure supplement 1D*).

## Macrophage-specific C3aR1 deletion does not alter glucose homeostasis

Owing to higher levels of *C3ar1* in murine MASLD and the differential regulation of *C3AR1* gene in MASLD humans, this motivated us to interrogate the role of pathophysiological role of *C3ar1* in macrophages in MASLD. We generated transgenic mice with macrophage-specific deletion of C3aR1 by crossing *C3ar1*^*flox/flox*^ mice with *Lyz2*^*Cre*^ transgenic mice (C3aR1-MφKO) to target both liver resident macrophages and recruited monocytes. *C3ar1*^*flox/flox*^ mice were used as controls. Successful deletion of *C3ar1* in macrophages from the C3aR1-MφKO mouse was confirmed by quantitative RT-PCR of isolated peritoneal macrophages that were F4/80+and CD68+by fluorescence-activated cell sorting (*Figure 2A*). In liver tissue, *C3ar1* expression was reduced by ~88% in both male and female C3aR1-MφKO (*Figure 2B*). These results indicate that macrophages account for the vast majority of *C3ar1* expression in the liver.

When placed on GAN diet, there was no significant difference in weight gain between control and C3aR1-MφKO mice (*Figure 2C*). There was similarly no difference in percent lean or fat mass between these mice (*Figure 2D*). Glucose tolerance tests performed in fasted mice after 27 weeks GAN diet found no significant differences between control and C3aR1-MφKO mice (*Figure 2E*). There was also no difference in insulin sensitivity as measured by insulin tolerance tests in male mice (*Figure 2— figure supplement 1A*). Insulin resistance as measured by comparing the ratio of fasting glucose level to fasting insulin level (HOMA-IR) was also unchanged between controls and C3aR1-MφKO mice (*Figure 2—figure supplement 1B*). Circulating serum ALT levels were unchanged in male control and C3aR1-MφKO mice on GAN diet (*Figure 2—figure supplement 1C*).

## Macrophage-specific C3aR1 deletion does not significantly impact hepatic steatosis or fibrosis

Liver samples collected after 28–30 weeks of GAN or regular diet did not show significant differences in liver mass between control and C3aR1-MφKO mice (*Figure 2F*). Male mice on GAN diet developed similar qualitative appearance on histology (*Figure 2G*), and slide image analysis showed similar proportions of lipid droplet area and collagen area (*Figure 2H and I*). This indicates that there were no significant differences in steatosis or fibrosis between GAN-fed control and C3aR1-MφKO male mice. While *C3ar1* expression was markedly reduced in the C3aR1-MφKO liver tissue (*Figure 2B*), there were no detectable gene expression changes in markers of fibrosis, inflammation, or lipid handling on

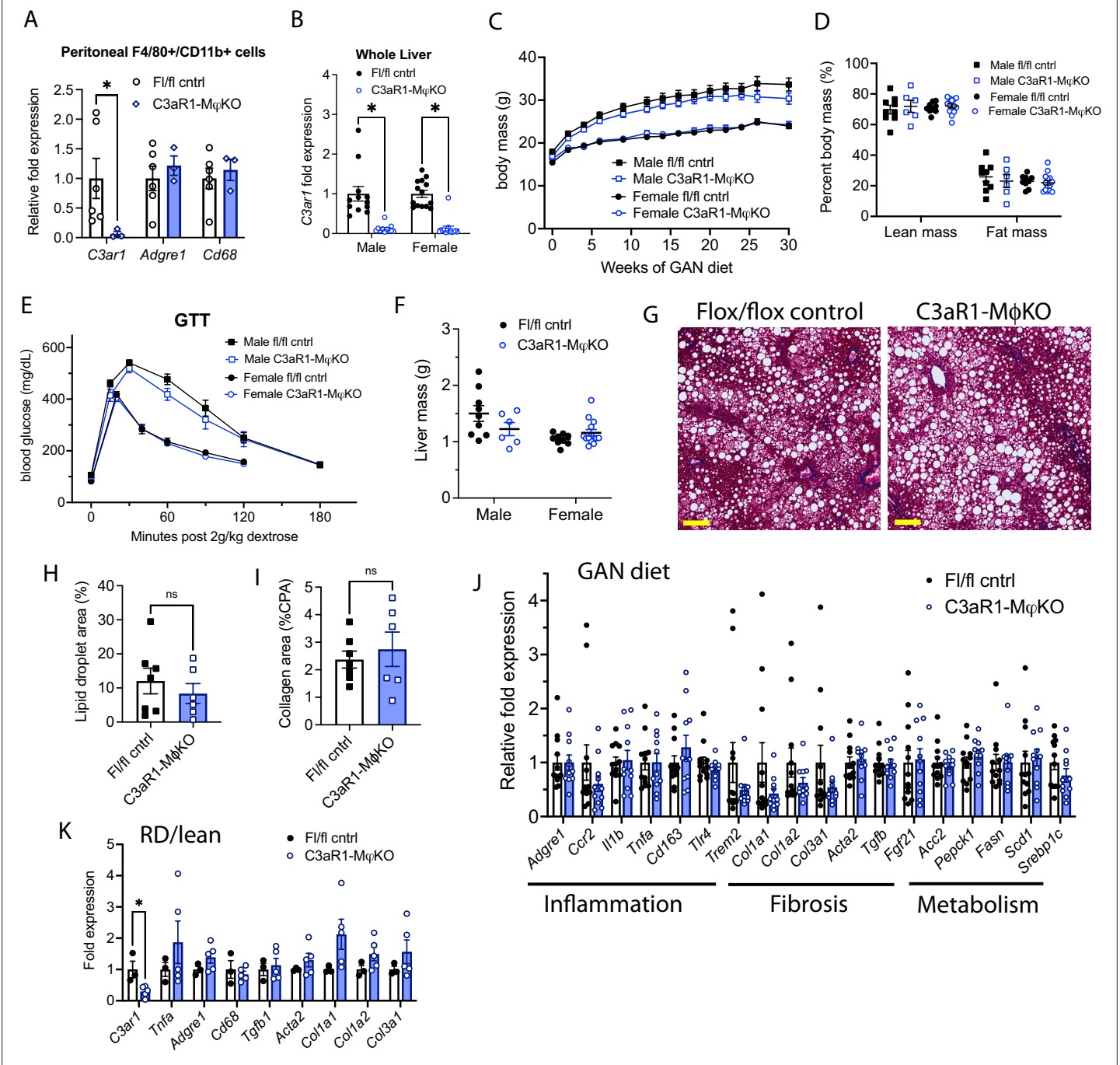

**Figure 2.** C3aR1 deletion in all macrophages does not affect weight gain, glucose homeostasis, liver steatosis or fibrosis. (**A**) Expression of *C3ar1* in peritoneal F4/80+/CD68+ cells from *C3ar1^flox/flox* control (n=6) or C3aR1-MφKO male mice (n=3). (**B**) Expression of *C3ar1* in whole liver from control or C3aR1-MφKO mice (n=11–12 per male group, n=13–14 per female group). (**C**) Body mass curve of control or C3aR1-MφKO mice on GAN high-fat diet starting at 5 weeks of age (n=11–12 per male group, n=14 per female group). (**D**) Body composition analysis by EchoMRI in control or C3aR1-MφKO mice after 30 weeks GAN diet (n=6–9 per male group, n=9–13 per female group). (**E**) Glucose tolerance test in control or C3aR1-MφKO mice with 14 hr fast after 28 weeks GAN diet (n=6–9 per male group, n=9–14 per female group). (**F**) Liver mass in control or C3aR1-MφKO male mice at time of euthanasia after 30 weeks GAN diet (n=6–9 per male group, n=9–14 per female group). (**H**) Representative liver section staining by Masson's Trichrome in male control or C3aR1-MφKO mice (scale bar = 100 mm). (**I**) Lipid droplet area in liver sections from male control or C3aR1-MφKO mice, excluding vessel lumens (n=6–7 per group). (**J**) Collagen area in liver sections from male control or C3aR1-MφKO mice (n=6–7 per group). (**J,K**) Relative mRNA expression of key markers for inflammation, fibrosis, and liver metabolism in liver from male control or C3aR1-MφKO mice after 30 weeks of either GAN (**J**) diet (n=11–12 per group) or regular (**K**) diet (n=3–5 per group). Unpaired two-tailed Student's *t* test: Student's *t* test: *, p<0.05. Error bars represent standard error of the mean.

*Figure 2 continued on next page*

*Figure 2 continued*

The online version of this article includes the following source data and figure supplement(s) for figure 2:

**Source data 1.** Source data for *Figure 2A-F, H-K*.

**Figure supplement 1.** C3aR1 deletion in macrophages does not affect insulin-glucose axis, circulating marker of liver injury, or expression of key genes in female mice.

either GAN or regular diet (*Figure 2J and K*). Similarly, in female mice there were also no significant differences between control and C3aR1-MφKO mouse liver on either GAN or regular diet in a subset of key gene markers of fibrosis or inflammation (*Figure 2—figure supplement 1D*).

## Kupffer-cell-specific C3aR1 deletion does not alter weight gain or glucose homeostasis

To explore whether there may be competing effects between recruited monocytes and liver resident macrophages (Kupffer cells), we next generated Kupffer-cell-specific C3aR1 knockout mice (C3aR1-KpKO) by crossing *C3ar1^flox/flox* mice to Clec4f-Cre transgenic mice and fed them GAN diet. *C3ar1^flox/flox* mice were used as controls. Body weight gain was similar between genotypes for both male and female mice (*Figure 3A*), and there was no difference in body composition between control and C3aR1-KpKO mice on GAN diet (*Figure 3B*). There was similarly no significant difference in glucose homeostasis between the genotypes during a glucose tolerance test (*Figure 3C*).

## Kupffer-cell-specific C3aR1 deletion does not significantly impact hepatic steatosis or fibrosis

Liver mass was not significantly different between control and C3aR1-KpKO mice on GAN diet (*Figure 3D*). Liver sections appeared qualitatively similar by histology stained with Masson's trichrome (*Figure 3E*). There were similar levels of hepatic steatosis in these mice as measured by percent lipid droplet area (*Figure 3F*). When measured by collagen proportional area, there was no significant differences in liver fibrosis between C3aR1-KpKO and control mice (*Figure 3G*). While *C3ar1* expression was reduced by 73% in liver tissue of C3aR1-KpKO mice, there were no significant differences in expression of inflammatory, fibrotic, or lipid handling gene markers (*Figure 3H*). *C3ar1* expression similarly decreased by ~90% in liver tissue of female C3aR1-KpKO mice fed regular diet compared to control mice (*Figure 3—figure supplement 1A*). These data also indicate that Kupffer cells account for ~80% of hepatic *C3ar1* gene expression in our mouse model of MASLD/MASH.

## Discussion

Overall, we found that macrophage or Kupffer cell expression of *C3ar1* does not impact body weight gain or histologic/transcriptomic features of MASLD/MASH in a murine dietary model. Deletion of C3aR1 in the macrophage population throughout the body, or specifically in Kupffer cells, did not affect weight gain, glucose homeostasis, or extent of hepatic steatosis/fibrosis. With long term GAN diet feeding that has been previously shown to model human MASLD/MASH, we did not observe significant differences in liver abnormalities with the KO mice.

Our findings in macrophage-specific C3aR1 KO mice contrast with prior observations in whole-body C3aR1 KO mice (*Mamane et al., 2009*), which are protected from diet-induced obesity, have improved glucose tolerance, and exhibit decreased hepatic steatosis. In both our macrophage- and Kupffer-cell-specific C3aR1 KO mice, which had similar degrees of obesity compared to controls, there was no detectable effect on liver steatosis or fibrosis despite the near abrogation of *C3ar1* expression. This raises the possibility that the lower levels of hepatic steatosis and insulin resistance previously observed in the whole body C3aR1 KO mice may be secondary to protection from obesity. Protection from diet-induced obesity in whole-body C3aR1 KO mice may be mediated by a non-macrophage cell type, since our macrophage-specific C3aR1 KO mice were not afforded this protection. The *C3ar1*-expressing cell types that promote obesity and MASLD remains to be determined.

Our laboratory recently reported sex-dependent regulation of thermogenic adipose tissue mediated by adipocyte-derived C3aR1 (*Ma et al., 2024*). However, no such sexual dimorphism was observed in hepatic expression of key MASH genes in response to GAN diet in our macrophage- or

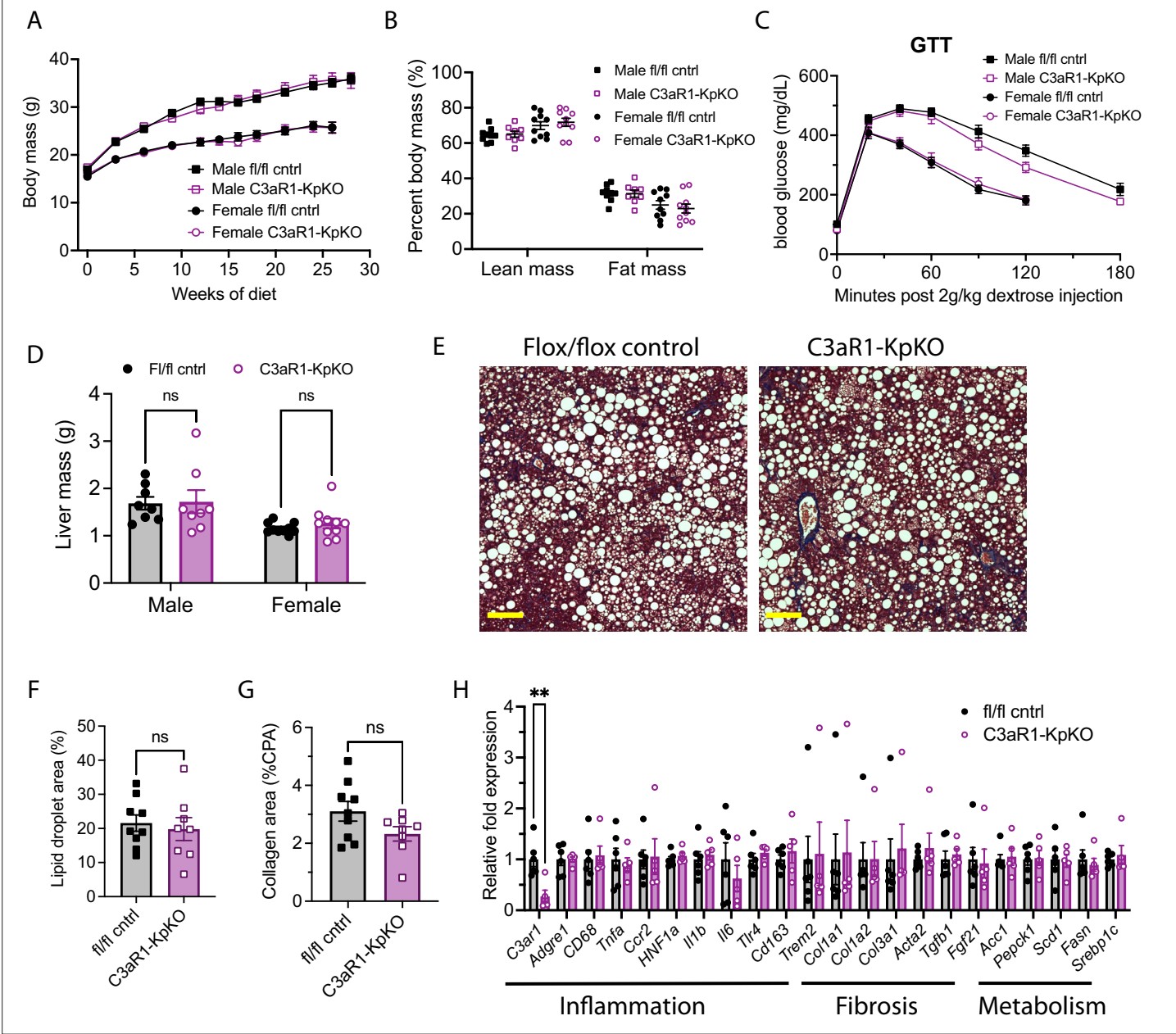

**Figure 3.** C3aR1 deletion in Kupffer cells does not affect weight gain, glucose homeostasis, liver steatosis or fibrosis. (**A**) Body mass curve on GAN diet in *C3ar1flox/flox* control or C3aR1-KpKO mice beginning at 5 weeks of age (n=8–10 per group). (**B**) Body composition analysis by EchoMRI in control or C3aR1-KpKO mice after 28 weeks GAN diet (n=8–10). (**C**) Glucose tolerance test in control or C3aR1-KpKO mice with 14 hr fast after 26 weeks GAN diet (n=8–10). (**D**) Liver mass in control or C3aR1-KpKO male mice at time of euthanasia after 30 weeks GAN diet (n=8–10). (**E**) Representative liver section staining by Masson's Trichrome in control or C3aR1-KpKO male mice (scale bar = 100 mm). (**F**) Lipid droplet area quantified on liver sections of control or C3aR1-KpKO male mice, excluding vessel lumens (n=8–9). (**G**) Collagen area quantified on whole liver section of control or C3aR1-KpKO male mice (n=8–9). (**H**) Relative gene expression in male control or C3aR1-KpKO mice after 30 weeks GAN diet (n=5–6). Unpaired two-tailed Student's *t* test: **, p<0.01. Error bars represent standard error of the mean.

The online version of this article includes the following source data and figure supplement(s) for figure 3:

**Source data 1.** Source data for *Figure 3A–D and F–H*.

**Figure supplement 1.** C3ar1 expression in female mice with Kupffer cell-specific deletion of *C3ar1*.

Kupffer-cell-specific C3aR1-deficient mice. Other work has suggested possible compensatory effects from its sister anaphylatoxin receptor C5aR1, with increased cold-induced adipocyte browning and attenuated diet-induced obesity seen in C3aR1/C5aR1 double KO mice (*Kong et al., 2023*).

The strengths of our study include careful metabolic and transcriptomic phenotyping of cell-type-specific transgenic mice. Some limitations were our use of a single MASLD dietary model and our focus on the C3aR1 pathway. While the GAN diet recapitulates many features of human MASH due to its similarity to Western diet (*Vacca et al., 2024*), relatively low levels of fibrosis were seen in our study, potentially related to initiating the diet at young age; more rapid fibrosis induction has been seen when GAN diet is initiated at older ages (*Li et al., 2023*). It is possible that in other models of liver injury that we did not test (e.g. short-term treatment with a hepatotoxin such as carbon tetrachloride; *Tsuchida et al., 2018*) there may be differences in liver injury in mice lacking *C3ar1* in macrophages. However, the GAN diet model has been shown to better parallel the gene expression changes in human MAFLD/MASH (*Hansen et al., 2020*). Lastly, while *C3AR1/C3ar1* expression is very low in non-macrophage cells (*Figure 1B*), C3aR1 signaling on other hepatic cell types not explored in this study, such as hepatic stellate cells, could mediate the observed effect in the whole-body C3aR1 KO mouse.

Deletion of C3aR1 in macrophages generally, or in liver resident macrophages specifically, had no major effect on systemic glucose homeostasis and hepatic steatosis, inflammation, and fibrosis in this murine dietary model of MASLD/MASH. The complement system is a complex entity directing an important part of the body's inflammatory and tissue repair response in MASLD. Further work is needed to elucidate the mechanisms of the role of C3aR1 in the pathogenesis of MASH and cirrhosis.

## Materials and methods

### Animals

*C3ar1^flox/flox* mice were on the C57BL/6 J background as described (*Cumpelik et al., 2021*). Homozygous *Lyz2^Cre* mice on the C57BL/6 J background (Strain #004781) as well as homozygous Clec4f-Cre mice on the C57BL/6 J background (Strain #003296) were purchased from Jackson Laboratories. *C3ar1^flox/flox* homozygous mice on C57BL/6 J background were used in the experiments as controls from the same backcross generation (*Ma et al., 2024*). All mice were maintained in plastic cages under a 12 hr/12 hr light/dark cycle at constant temperature (22 °C) with free access to water and food. Mice were fed regular diet containing 4.5%kcal fat PicoLab Rodent diet 20 (LabDiet) or GAN diet containing 40%kcal HFD (mostly palm oil) with 20% fructose and 2% cholesterol (D09100310, Research Diets) for 28–30 weeks. Fat mass and lean mass were determined via noninvasive 3-in-1 body composition analyzer (EchoMRI). Mice were humanely euthanized with $CO_2$ inhalation followed by exsanguination by cardiac puncture. For a typical experiment we expected ~10% loss of animals, a coefficient of variation (CV) of 10% and a treatment/genotype effect of 30–50%. To ensure an adequate statistical power of 0.9 with an alpha value of 0.05, we anticipated 6–12 mice per experimental group for physiology experiments. Key experiments were repeated in at least two independent mouse cohorts.

### Blood chemistry and serum insulin analysis

Mice were fasted overnight (14–16 hr) for glucose tolerance tests and injected intraperitoneally with syringe-filtered D-glucose solution (2 g/kg). For insulin tolerance test, mice were fasted for 6 hr and injected with 0.5 mIU/kg insulin. Blood glucose levels were assayed by commercial glucometer (OneTouch) by tail vein blood samples. Plasma insulin levels were measured from mice fasted for 6 hr. Tail vein blood was collected into lithium heparin-coated tubes, centrifuged at 2000 x *g* at 4 °C, and plasma insulin levels were determined by ELISA using a standard curve (Mercodia). Serum alanine aminotransferase levels were measured in serum from blood collected via cardiac puncture using a commercially available colorimetric assay (TR71121, Thermo Fisher Scientific).

### Peritoneal macrophage isolation and flow cytometry

Peritoneal macrophages were isolated from as previously described (*Zhang et al., 2008*). Briefly, mice were euthanized then immediately injected intraperitoneally with 10mL phosphate-buffered saline (PBS, pH 7.4) at room temperature. After a 3–5min incubation period, peritoneal fluid was removed with sterile needle and syringe and placed on ice. After centrifugation at 300 x *g*, the pellet was resuspended in PBS containing 2% fetal bovine serum and 0.1% sodium azide. Cells were stained

with phycoerythrin-conjugated anti-F4/80 (clone BM8, cat. #123110) and fluorescein isothiocyanate-conjugated anti-CD11b (clone M1/70, cat. #101206) fluorescent antibodies (Biolegend). Stained cells were loaded on MA900 fluorescence-activated cell sorter (Sony), and dual-positive F480+/CD11b+ cells were sorted for subsequent RNA extraction.

## Histological studies

A mid-distal portion of the left liver lobe was fixed with 10% buffered formalin and transferred to 70% ethanol. Samples were embedded in paraffin, sectioned at ~5 µm thickness, and stained with Masson's trichrome. Slides were imaged using Zeiss Axioscan7 at ×20 magnification. Histologic analyses were performed using ImageJ software (version 1.53t). Lipid droplet area was quantified by subtracting non-droplet area in the green channel from total section area of two to three independent sections. Collagen proportionate area was quantified by measuring total area in the red channel after reducing intensity threshold to 60–70.

## RNA extraction and real-time quantitative PCR analysis

Total RNA from liver tissue lysates was extracted using Trizol reagent (Invitrogen) followed by RNAeasy Mini kit (QIAGEN) as per manufacturer's protocol. RNA was reverse-transcribed using the High Capacity cDNA RT kit (Thermo Fisher). Quantitative PCR was performed using SYBR Green Master Mix (Quanta) and specific gene primers on QuantStudio6 Flex Real-Time PCR Systems (Thermo Fisher Scientific) using the delta-delta Ct method. Expression levels were normalized to Ribosomal protein S18 (*Rps18*). Primer sequences are listed in *Supplementary file 1*.

## Statistical analyses

All statistical analyses were performed with biological replicates using GraphPad Prism10. Unpaired two-tailed Student's *t* test with Welch correction for most analyses, with Holm-Šídák correction for multiple comparisons where applicable, and $p < 0.05$ was considered statistically significant.

## Acknowledgements

We would like to thank Dr. Baran Ersoy, Dr. Robert Schwartz, and Dr. Saloni Sinha for their technical advice and assistance. EAH was supported by NIH T32 5T32HL160520-02. AG was supported by ADA 9–22-PDFPM-01. RPL was supported by AHA 23DIVSUP1074485. LS was supported by AHA 908952 and an Ehrenkranz Young Scientist Award. JCL was supported by NIH R01 DK121140, R01 DK121844, and R01 DK132879. The views expressed in this manuscript are those of the authors and do not necessarily represent the official views of the American Diabetes Association, the American Heart Association, the National Institute of Diabetes and Digestive and Kidney Diseases, or the National Institutes of Health.

## Additional information

### Funding

| Funder | Grant reference number | Author |
| --- | --- | --- |
| National Institutes of Health | DK121140 | James C Lo |
| National Institutes of Health | DK121844 | James C Lo |
| National Institutes of Health | DK132879 | James C Lo |
| National Institutes of Health | 5T32HL160520 | Edwin A Homan |
| American Diabetes Association | 9-22-PDFPM-01 | Ankit Gilani |

| Funder | Grant reference number | Author |
|--------|------------------------|--------|
| American Heart Association | 10.58275/aha. 23divsup1074485.pc.gr. 168377 | Renan Pereira de Lima |
| American Heart Association | 908952 | Lisa Stoll |

The funders had no role in study design, data collection and interpretation, or the decision to submit the work for publication.

## Author contributions

Edwin A Homan, Formal analysis, Funding acquisition, Investigation, Writing – original draft, Project administration, Writing – review and editing; Ankit Gilani, Formal analysis, Investigation, Writing – review and editing; Alfonso Rubio-Navarro, Lisa Stoll, Investigation, Writing – review and editing; Maya A Johnson, Odin M Schaepkens, Eric Cortada, Renan Pereira de Lima, Investigation; James C Lo, Conceptualization, Supervision, Funding acquisition, Writing – original draft, Writing – review and editing

## Author ORCIDs

Edwin A Homan (iD) https://orcid.org/0000-0002-2923-9635
James C Lo (iD) https://orcid.org/0000-0003-0244-1670

## Ethics

This study was performed in strict accordance with the recommendations in the Guide for the Care and Use of Laboratory Animals of the National Institutes of Health. All of the animals were handled according to approved institutional animal care and use committee (IACUC) protocols of the Weill Cornell Medical College. The protocol was approved by the Committee on the Ethics of Animal Experiments of the Weill Cornell Medical College (protocol#: 2015-0020).

Reviewer #1 (Public review): https://doi.org/10.7554/eLife.100708.3.sa1
Reviewer #2 (Public review): https://doi.org/10.7554/eLife.100708.3.sa2
Author response https://doi.org/10.7554/eLife.100708.3.sa3

# Additional files

## Supplementary files

Supplementary file 1. Table containing forward and reverse primer sequences for the gene targets used in quantitative PCR experiments performed in this study.

MDAR checklist

## Data availability

All data generated or analyzed during this study are included in the manuscript and supporting files; source data files, where applicable, have been provided for Figures 1, 2, and 3, as well as associated figure supplements. Figure 1A was analyzed from the previously published dataset by *Uhlén et al., 2015*. Figure 1B was analyzed from the previously published dataset by *MacParland et al., 2018*. Figure 1C was analyzed from previously published dataset *Suppli et al., 2019*. Figure 1 - figure supplement 1A was analyzed from previously published dataset *The Tabula Muris Consortium et al., 2018*.

The following previously published datasets were used:

| Author(s) | Year | Dataset title | Dataset URL | Database and Identifier |
|---|---|---|---|---|
| Suppli MP, Rigbolt KTG, Veidal SS, Heebøll S, Eriksen PL, Demant M, Bagger JI, Nielsen JC, Oró D, Thrane SW, Lund A, Strandberg C, Kønig MJ, Vilsbøll T, Vrang N, Thomsen KL, Grønbæk H, Jelsing J, Hansen HH, Knop FK | 2019 | Hepatic transcriptome signatures in patients with varying degrees of nonalcoholic fatty liver disease compared with healthy normal-weight individuals | https://www.ncbi.nlm.nih.gov/geo/query/acc.cgi?acc=GSE126848 | NCBI Gene Expression Omnibus, GSE126848 |
| The Tabula Muris Consortium | 2018 | Tabula Muris: Transcriptomic characterization of 20 organs and tissues from Mus musculus at single cell resolution | https://www.ncbi.nlm.nih.gov/geo/query/acc.cgi?acc=GSE109774 | NCBI Gene Expression Omnibus, GSE109774 |

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
