## [Editor Report · eLife Assessment]

This **valuable** study investigates the role of Complement 3a Receptor 1 (C3aR) in the pathogenesis of Metabolic Dysfunction-Associated Steatotic Liver Disease (MASLD) using mouse models with specific target deletions in various cell types. While the general relevance of C3aR in inflammatory contexts has been established before, the authors provide **solid** evidence here that C3aR does not contribute significantly to MASLD pathogenesis in their models. The work will be of interest to colleagues studying diseases of the liver and the intersection with inflammation.

---

## [Referee Report · Reviewer #1 (Public review)]

Summary:

In this paper Homan et al used mouse models of Metabolic Dysfunction-Associated Steatotic Liver Disease and different specific target deletions in cells to rule out the role of Complement 3a Receptor 1 in the pathogenesis of disease. They provided limited evidence and only descriptive results that despite C3aR being relevant in different contexts of inflammation, however, these tenets did not hold true.

Comments on revisions:

The revised version fulfilled my queries.

---

## [Referee Report · Reviewer #2 (Public review)]

Summary:

Homan et al. examined the effect of macrophage- or Kupffer cell-specific C3aR1 KO on MASLD/MASH-related metabolic or liver phenotypes.

Strengths:

Established macrophage- or Kupffer cell-specific C3aR1 KO mice, and showing comparable liver metabolic phenotypes between WT and macrophage-specific C3aR1KO mice in response to normal chow diet or MASH diet feeding.

Weaknesses:

Insufficient data showing the effects of C3aR1KO on liver macrophage phenotypes, such as hepatic macrophage profiles, macrophage activation status, etc, which are important for the development of liver steatosis and fibrosis.

---

## [Author Response]

The following is the authors’ response to the original reviews.

**Public Reviews:**

**Reviewer #1 (Public review):**
Summary:In this paper Homan et al used mouse models of Metabolic Dysfunction-Associated Steatotic Liver Disease and different specific target deletions in cells to rule out the role of Complement 3a Receptor 1 in the pathogenesis of disease. They provided limited evidence and only descriptive results that despite C3aR being relevant in different contexts of inflammation, however, these tenets did not hold true.Weaknesses:(1) The results are based on readouts showing that C3aR is not involved in the pathogenesis of liver metabolic disease.(2) The description of the mouse models they used to validate their findings is not clear. Lysm-cre mice - which are claimed to delete C3aR in (?) macrophages are not specific for these cells, and the genetic strategy to delete C3aR in Kupffer cells is not clear.(3) Taking this into account, it is very challenging to determine the validity of these data, also considering that they are merely descriptive and correlative.

We generated 2 different cohorts of mice using LysM-Cre (Jackson Strain #004781) to drive deletion in all macrophages and Clec4f-Cre (Jackson Strain #033296) to specifically ablate *C3ar1* in Kupffer cells. These experimental models have been clearly defined in the revised manuscript on pages 5 and 7 and in the methods section (page 10). The reviewer’s point is well taken that the LysM-Cre transgene can also be active in granulocytes and some dendritic cells. Even so, despite deletion of *C3ar1* in macrophages and other granulocytes, we do not see a major effect on hepatic steatosis and fibrosis in this GAN diet induced model of MASLD/MASH. This was a somewhat surprising finding. We do not agree that our findings are correlative. We specifically ablated C3aR1 in macrophages or Kupffer cells and found no significant differences in the major readouts of steatosis and fibrosis for MASLD/MASH between control and knockout mice. It is possible that in other models of liver injury that we did not test (e.g., short-term treatment with a hepatotoxin such as carbon tetrachloride), there may be differences in liver injury in mice lacking *C3ar1* in macrophages, but the GAN diet model has been shown to better parallel the gene expression changes in human MAFLD/MASH. This has been added to the discussion (page 9).

**Reviewer #2 (Public review):**
Summary:Homan et al. examined the effect of macrophage- or Kupffer cell-specific C3aR1 KO on MASLD/MASH-related metabolic or liver phenotypes.Strengths:Established macrophage- or Kupffer cell-specific C3aR1 KO mice.Weaknesses:Lack of in-depth study; flaws in comparisons between KC-specific C3aR1KO and WT in the context of MASLD/MASH, because MASLD/MASH WT mice likely have a low abundance of C3aR1 on KCs.Homan et al. reported a set of observation data from macrophage or Kupffer cell-specific C3aR1KO mice. Several questions and concerns as follows could challenge the conclusions of this study:(1) As C3aR1 is robustly repressed in MASLD or MASH liver, GAN feeding likely reduced C3aR1 abundance in the liver of WT mice. Thus, it is not surprising that there were no significant differences in liver phenotypes between WT vs. C3aR1KO mice after prolonged GAN diet feeding. It would give more significance to the study if restoring C3aR1 abundance in KCs in the context of MASLD/MASH.

GAN diet feeding resulted in higher liver *C3ar1* compared to regular diet (Figure 1H). This thus became an impetus for studying the effects of *C3ar1* deletion in macrophages or Kupffer cells, which are responsible for the majority of liver *C3ar1* expression, in MASLD/MASH (Figures 2B and 3H). This point has been added to the text on page 5.

(2) Would C3aR1KO mice develop liver abnormalities after a short period of GAN diet feeding?

We did not assess if short term GAN diet feeding resulted in significant differences in liver abnormalities in the *C3ar1* macrophage or Kupffer cell knockout mice. Perhaps the reviewer’s point is that perhaps with shorter periods of GAN diet feeding there may be a phenotype in the KO mice. We agree that this is entirely possible, though with shorter feeding timeframes what is typically seen is hepatic steatosis without fibrosis. Nevertheless, the most important element in our opinion for a disease preventing or modifying model lies with the longer-term GAN diet feeding. With long term GAN diet feeding that has been previously shown to model human MASLD/MASH, we did not observe significant differences in liver abnormalities with the KO mice. This has been added to the discussion (page 8).

(3) What would be the liver macrophage phenotypes in WT vs C3aR1KO mice after GAN feeding?

Similar to the above point, given the lack of a major MASLD/MASH phenotype in hepatic steatosis and fibrosis, we did not further profile the liver macrophage profiles of the macrophage or Kupffer cell *C3ar1* KO mice with GAN feeding.

(4) In Fig 1D, >25wks GAN feeding had minimal effects on female body weight gain. These GAN-fed female mice also develop NASLD/MASH liver abnormalities?

We thank the reviewer for this question. In general, female GAN-fed mice develop milder MASLD/MASH abnormalities. We have included additional data in the revised manuscript in Figure S4. These results show no to minimal development of a MASLD/MASH gene signature.

(5) Would C3aR1KO result in differences in liver phenotypes, including macrophage population/activation, liver inflammation, lipogenesis, in lean mice?

We have provided additional data further characterizing liver inflammation, lipogenesis and macrophages in macrophage *C3ar1* KO mice under lean/regular diet conditions in Figure 2K. These results show a potential trend but no substantial development of a MASLD/MASH gene signature.

(6) The authors should provide more information regarding the generation of KC-specific C3aR1KO. Which Cre mice were used to breed with C3aR1 flox mice?

Clec4f-Cre transgenic mice were used to generate Kupffer cell specific KO of *C3ar1*. This has been clarified and explicitly stated in the revised manuscript on page 7 and in the methods section.

**Recommendations for the authors:**

**Reviewer #1 (Recommendations for the authors):**
These data should be repeated using a more established model of Kupffer cell target deletion via Clec4-F mice.

Our data with Kupffer cell *C3ar1* deletion is indeed done with Clec4f-Cre transgenic mice. This has been clarified in the revised manuscript on page 7 and in the methods section.

**Reviewer #2 (Recommendations for the authors):**
(1) Typo: "iver" in the abstract(2) Line 97, "GAN diet I" should be "GAN diet"?

These points have been corrected in the revised manuscript.